# Analysis and Optimization of a Microgripper Driven by Linear Ultrasonic Motors

**DOI:** 10.3390/mi13091453

**Published:** 2022-09-02

**Authors:** Ranran Geng, Zhiyuan Yao, Yuqi Wang, Jiacai Huang, Hanzhong Liu

**Affiliations:** 1Industrial Center, Advanced Industrial Technology Research Institute, Nanjing Institute of Technology, Nanjing 211167, China; 2State Key Laboratory of Mechanics and Control of Mechanical Structures, Nanjing University of Aeronautics and Astronautics, Nanjing 210016, China

**Keywords:** microgripper, vibration analysis, optimal design, sensitivity calculation, linear ultrasonic motors

## Abstract

This paper presents the vibration response analysis and optimal structural design of a microgripper driven by linear ultrasonic motors (LUMs) dedicated to improving end-point positioning accuracy. Based on structural vibration theory, a parametric vibration response model of the microgripper finger was established, and the relative sensitivities of the structural and material parameters that affect the vibration amplitude of the fingertip were calculated within the structural and material constraints. Then, according to the sensitivity calculation results, a multidimensional constrained nonlinear optimization model was constructed to suppress the vibration of the end-effector. The improved internal penalty function method combined with Newton iteration was adopted to obtain the optimal structural parameters. Finally, the vibration experimental results show that the vibration amplitude of the initial microgripper fingertip is 16.31 μm, and the value measured after optimization was 2.49 μm, exhibiting a reduction of 84.7%. Therefore, the proposed optimal design method can effectively restrain the vibration of the microgripper end-effector and improve manipulation stability.

## 1. Introduction

In recent decades, with the rapid development of micro and nanotechnology and the continuous miniaturization of operating instruments, microoperation technology has attracted much attention [1,2,3,4]. Microgripping is one of the most important branches of microoperation technology, which involves many fields of discipline, such as the processing, modification, and inspection of micronano parts in the mechanical engineering field; assembly operations in micro-electromechanical systems (MEMS); and the high-precision operation of biological cells in the biomedical field [5,6,7,8]. With the development of bioengineering and medical science, substantial demands have been placed on the dynamic properties and manipulation performances of microgrippers. The microgripper is required to have a high driving velocity, precision, and payload capacity; be light weight; have low power consumption; and provide great performance flexibility. However, lighter manipulators are usually characterized by low rigidity; this can easily cause vibrations during actual operation, thereby affecting the positioning accuracy and repeatability of microgrippers in high-speed engineering applications.

Nowadays, many scholars are concentrating on vibration suppression research for microgrippers [9,10,11,12]. An input shaping algorithm was proposed as a method of suppression control for residual vibrations of a PZT gripper caused by its flexible structure [13]. With the control method, the residual vibration level of the PZT gripper was reduced to 10% by using a noncontact laser Doppler vibrometer. Yang presented a hybrid control scheme to simultaneously suppress the elastic vibrations of a flexible micromanipulator bonded with one macro-fiber composite actuator [14]. In the study, two optimization indices regarding the comprehensive torques and synthesized vibrations were designed, and fuzzy variable structure control with nonlinear adaptive control law was created. The experiments show that the excited vibrations during the motor motion and the residual vibration after the motor motion were decreased, which verified the effectiveness and feasibility of the established system model. Ejima and Ohara developed a two-fingered microhand which uses a parallel mechanism to solve problems about workspace and vibrations [15]. The vibration analysis through a simulated transportation task indicated the vibration amplitude of the end-effector was 4.64 μm. Yavuz considered a single-link, flexible composite manipulator to analyze ANSYS and reduce end-point vibrations [16]. The trapezoidal and triangular velocity profiles were studied for motion control commands, and the results show that the trapezoidal mode has a good inhibitory effect on the residual amplitude.

Vibration control methods can generally suppress the residual vibration of the microgripper end-effector in the field of mechanical engineering [17,18,19,20]. However, as the microgripper shifts from the rigid model to the flexible model, the end-effector positioning problem caused by vibrations will become central. It is difficult to control the position of microgrippers through signal control due to their complex structure and multimodal input. Therefore, determining how to inhibit the vibrations of microgrippers has become an urgent problem.

The microgripper studied in this paper is driven by linear ultrasonic motors (LUMs), which can improve the response ability and operating efficiency due to their high positioning accuracy and self-locking feature [21,22,23,24,25]. However, LUMs also have characteristics such as large instantaneous output force and quick action time [26]. These characteristics cause the vibrations of the microgripper to become difficult to control. Therefore, in this paper, the vibration of the fingertip of the microgripper is the main research object. A parameterized model of the microgripper was developed, and the sensitivity of each input parameter was analyzed based on the model. Furthermore, according to the results of a sensitivity analysis, a vibration optimization model of the microgripper was established, and the interior penalty function method and Newton method were used to solve the equations of the optimization model. Finally, the experimental results demonstrate that the vibration of the fingertip of the microgripper was effectively inhibited, thereby verifying the feasibility of the structural optimization model and improving the stability and practicability of the microgripper.

## 2. Vibration Model

### 2.1. Structure of Microgripper

Parallel mechanisms have been widely applied in the structural design of the microgrippers, as they possess the advantages of high payload ratio, high accuracy, better stability, and no accumulative error [27,28,29,30]. The microgripper adopts a chopstick-like finger and spatial parallel mechanical design to make the mechanism more compact, simplified, and easier to operate. Here we provide a brief summary of the design of the micro-manipulator for clarity. Details of this design may be found in our previous papers [31,32].

The microgripper has a parallel double-layer structural form, as is shown in Figure 1. The lower layer is the base layer consisting of a fixed base and a static finger, and the upper layer is the moving layer consisting of a moving platform and a moving finger fixed on the platform. Three LUMs and their guide rails distributed on the base symmetrically. There are three flexible hinges connecting the guide rails and the moving platform. The direction of the linear movement of the guide rails driven by LUMs is perpendicular to the two layers. These three movements of the guide rails lead to the 3-DOF motion of the platform, including one translational direction and two rotational directions through the deformation of the flexible hinges.

The actuator used in the microgripper is V-type linear ultrasonic motor, shown in Figure 2, which has higher output force and more stable running characteristics.

### 2.2. Vibration Equation

During the process of driving the moving finger, the step distance of the LUM is very small, namely, within dozens of nanometers. Therefore, it can be assumed that the deformation of the moving platform is linear and elastic. The mechanical model of the upper layer is presented in Figure 3, in which the impulse generated from the LUM is *p*; the speed in the perpendicular direction to the axis of the finger is *v*; the mass of the moving finger base is *M*; the angle between the moving finger and the moving platform is *θ*; and the elastic modulus, moment of inertia, density, length, and diameter of the moving finger are *E*, *I*, *ρ*, *l*, and *d*, respectively.

A structure with a moving finger fixed at one end and free at the other is similar to a cantilever. After the LUM completes a stepping motion, the fixed end of the finger, namely, “*A*” in Figure 2, is fixed due to being self-locking. Thus, the vibration equation of the free end of the moving finger can be written as follows.
(1)$$w(x,t)=\sum_{n=1}^{+\infty }W_{n}(x)(b_{1n}\cos\omega_{n}t+b_{2n}\sin\omega_{n}t),$$
where *ω_n_* is the natural frequency of moving finger, *b*_1*n*_ and *b*_2*n*_ are undetermined parameters, and *W_n_*(*x*) is the natural mode function of moving finger, which can be expressed as:(2)$$W_{n}(x)=\cosh s_{n}x-\cos s_{n}x+v_{n}(\sinh s_{n}x-\sin s_{n}x),$$
(3)$$v_{n}=-\frac{\sinh s_{n}l-\sin s_{n}l}{\cosh s_{n}l+\cos s_{n}l}.$$

The natural frequency of moving finger can be written as:(4)$$\omega_{n}^{}={\left(s_{n}^{}l\right)}^{2}\sqrt{\frac{EI}{\rho Al^{4}}},$$
where *A* represents the cross-sectional area of the moving finger. By using Matlab software to solve Equation (5), $s_{n}l$ can be obtained:(5)$$\cos s_{n}l\cosh s_{n}l=-1,$$
(6)$$s_{n}l=\begin{array}{cc} 1.8751, & \begin{array}{cc} 4.6941, & \begin{array}{cc} 7.8548, & \begin{array}{cc} 10.9955, & \cdot \cdot \cdot \end{array} \end{array} \end{array} \end{array}$$

The initial condition of the vibration equation is:(7)$$\{\begin{array}{c} w(x,0)=0\\ \dot{w}(x,0)=v \end{array}(0\leq x\leq l).$$

Then, by substituting Equation (1) into (7), we can get:(8)$$b_{1n}=0,$$
(9)$$b_{2n}=\frac{v\int_{0}^{l}W_{n}(x)dx}{\omega_{n}\int_{0}^{l}W_{n}^{2}(x)dx}=\frac{v}{\omega_{n}}K,$$
where *K* is the constant associated with $s_{n}l$. According to the momentum theorem, the velocity of the moving finger when its base receives the impulse *p* can be expressed as:(10)$$v=\frac{2p\cos\theta }{2\rho Al+M}.$$

Finally, the vibration response of the moving finger free end can be written as:(11)$$w(l,t)=\sum_{n=1}^{+\infty }W_{n}(l)\frac{2pl^{2}\cos\theta }{(2\rho Al+M){\left(s_{n}l\right)}^{2}}\sqrt{\frac{\rho A}{EI}}K\sin\omega_{n}t,$$
in which $A=\frac{\pi d^{2}}{4}$, $I=\frac{\pi d^{4}}{64}$.

The constant in the vibration equation is removed, and the structural material parameters of the finger are taken as the objective function. The equation can be expressed as:(12)$$G(E,d,\rho ,l)=\frac{16l^{2}}{(\pi \rho d^{2}l+2M)d}\sqrt{\frac{\rho }{E}}.$$

### 2.3. Constraint Conditions

An extreme point may be in the feasible region in an optimal case, but it would be outside the feasible region in most engineering cases. Under these circumstances, the extreme points of unconstrained optimization do not coincide with the extreme points of constrained optimization; therefore, constrained optimization is more complicated than unconstrained optimization. Excluding the objective function itself, the solution of constrained optimization is also related to the properties of the objective function. For the optimization of the finger vibration of the microgripper, the constraints are primarily geometric and material constraints.

#### 2.3.1. Material Constraints

The material constraints mainly include the elastic modulus *E* and the density *ρ* of the material, that is
(13)$$E^{\min}<E<E^{\max},$$
(14)$$\rho^{\min}<\rho <\rho^{\max},$$
where *E*^max^ and *E*^min^ represent the maximum and minimum values of the elastic modulus of the material, respectively; and *ρ*^max^ and *ρ*^min^ represent, respectively, the maximum and minimum values of the density of the material. As the elastic modulus and density are fixed, material constraints play an important role in engineering.

#### 2.3.2. Geometric Constraints

The design concept of the microgripper is based on the principle of humans using two chopsticks to grip an object. In practical applications, the length of the finger should be limited to within a certain range: if it is too short or too long, problems such as a weak grip and the easy falling of the object can occur, or an oversized mechanism can decrease the accuracy of the operation positioning. Moreover, the microgripper is used to grip tiny objects at the micro–nano level, so the gripping operation must be performed under a microscope. A too-thick or too-thin diameter of the finger will affect the difficulty of actual operation: if it is too thick, it will be difficult to focus under a microscope lens, and if it is too small, the finger vibration will be increased. Therefore, the diameter and length of the finger must be limited to within a certain range, and the geometric constraints are as follows:(15)$$d^{\min}<d<d^{\max},$$
(16)$$l^{\min}<l<l^{\max},$$
where *d*^min^ and *d*^max^ represent, respectively, the minimum and maximum cross-sectional diameters of the moving finger; and *l*^min^ and *l*^max^ represent, respectively, the minimum and maximum lengths of the moving finger. The ranges of the structural and material parameters of the finger are listed in Table 1.

### 2.4. Sensitivity Analysis

The geometric and material parameters of the microgripper structure are reflected in the physical parameters of the dynamic model. The sensitivities of the dynamic characteristics of the microgripper finger to the modifiable parameters can be calculated by mathematical methods [33]. There are two ways to define sensitivity. The first is the change in the dependent variable divided by the change of the independent variable, and the other is the relative change in the dependent variable divided by the relative change in the independent variable. To facilitate the comparison between the influences of various parameters on the dynamic characteristics of the finger, the latter definition is adopted in the present study.

Suppose the objective function is $G=G(x_{1},x_{2},x_{3},x_{4})$, where $x_{i}\in (E,d,\rho ,l)$, G∈R. Then,
(17)$$\eta (G/x_{i})=\underset{\Delta x_{i}\to 0}{\lim}\frac{\Delta G/G}{\Delta x_{i}/x_{i}}=\frac{x_{i}}{G}\cdot \frac{\partial G}{\partial x_{i}}(x_{i}\neq 0,G\neq 0),$$
which is defined as the relative sensitivity of *G* to *x_i_*. Figure 4 shows the relative sensitivity of the objective function to the elastic modulus *E*, density *ρ*, diameter *d*, and length *l* of the finger.

It can be seen in the figure that each parameter has a great influence on the value of the objective function. Among them, the elastic modulus *E* has the least effect on the vibration of the fingertip, and the diameter *d* has the greatest effect.

## 3. Structure Optimization

The Newton method and internal penalty function method can be combined to solve the problem of multidimensional constraint optimization in engineering. The internal penalty function method is a mathematical method by which to indirectly solve the multidimensional constrained optimization problem. It integrates the constraints in the optimization model into the objective function in a certain form, and the optimization problem with constraints is transformed into an unconstrained optimization problem. This method is quick and convenient for the suppression of the vibration of the moveable finger. However, a series of designs generated by the internal penalty function method requires the initial design results to be located in the feasible region. For complex problems, due to the approximation of the numerical calculation, the results may leave the feasible region after some iterations, thereby rendering the algorithm ineffective. Therefore, it is necessary to improve the internal penalty function method to obtain more accurate solutions in practical engineering.

### 3.1. Optimization Model

Reasonable design is the fundamental guarantee of the accurate operation of the microgripper. The properties of materials must be considered in the design, and the influence of the finger on the control of the microgripper should also be taken into account [34,35,36,37]. The multidimensional constraint equation of the moveable finger of the microgripper can be expressed as:(18)$$\begin{array}{l} \min G(E,d,\rho ,l)\\ s.t.\left\{ \begin{array}{l} g_{u1}:E^{\min}<E<E^{\max}\\ g_{u2}:d^{\min}<d<d^{\max}\\ g_{u3}:\rho^{\min}<\rho <\rho^{\max}\\ g_{u4}:l^{\min}<l<l^{\max} \end{array}\right. \end{array}$$
where *g_ui_*(*i =* 1, 2, 3, 4) is the boundary constraint condition. The internal penalty function method, which is the unified obstacle factor, is used to deal with the objective function. This method can be successful for the control of the variables within the feasible region because it has little error when the values of the design variables are close to each other; however, if the values vary greatly, the obstacle items 1/*g_ui_*(*x*) are usually not on the same order of magnitude, and the design variables cannot be guaranteed to be in the feasible region under the approximate calculation. Therefore, different obstacle factors can be used for different constraint conditions. The objective function that uses the improved internal penalty function method can be written as:(19)$$f(x,r_{i}^{k})=G(x)+\sum_{i=1}^{4}\frac{r_{i}^{k}}{g_{ui}(x)},$$
where x=[E,d,ρ,l], $r_{i}^{k}(i=1,2,3,4)$ is the obstacle factor of a decreasing sequence, and *k* is the number of iterations. There is:(20)$$r_{i}^{1}>r_{i}^{2}>r_{i}^{3}>r_{i}^{4}>...0,$$
and $\underset{k\to \infty }{\lim}r_{i}^{k}=0$. The initial obstacle factor $r_{i}^{0}$ is selected according to the empirical formula, which can be written as:(21)$$r_{i}^{0}=\frac{G(x^{0})}{\sum_{n=1}^{2}\frac{1}{g_{ui}(x^{0})}},$$
where $x^{0}$ is the initial iteration point, each obstacle term 1/*g_ui_*(*x*) contains two equations, and the obstacle factor $r_{i}^{k+1}=Cr_{i}^{k}$, *C* < 1 is the decreasing coefficient. In this way, the problem is transformed into a multidimensional unconstrained optimization problem, and Newton’s method is adopted for iterative solution. The iterative function is as follows:(22)$$x^{k+1}=x^{k}-\frac{f^{\prime }(x,r_{i}^{k})}{f^{\prime \prime }(x,r_{i}^{k})}.$$

According to the necessary conditions for the existence of extremum $\nabla f(x^{k+1},r_{i}^{k})=0$, the iterative function can be written as:(23)$$x^{k+1}=x^{k}-{\left[\nabla^{2}f(x^{k},r_{i}^{k})\right]}^{-1}\nabla f(x^{k},r_{i}^{k}),\;k=0,1,2,\cdot \cdot \cdot$$
where $\nabla^{2}f(x^{k},r_{i}^{k})$ is the Hessian matrix of the function $f(x^{k},r_{i}^{k})$ at point $\left[x^{k},r_{i}^{k}\right]$.

Theoretically, any iteration algorithm must converge to the optimal point after passing through infinite iterations; however, in practical engineering, there is no need to carry out infinite iterations. Generally, as long as the iteration point is close to the minimum point, the optimal point is considered to have been found. The criterion that can be judged to terminate the iterative calculation is the convergence criterion of the optimization. The convergence condition of the vibration of the moveable finger can adopt the decline criterion as follows:(24)$$\left|f(x^{k+1},r_{i}^{k+1})-f(x^{k},r_{i}^{k})\right|\leq \varepsilon ,$$
where *ε* is the convergence accuracy. The convergence accuracy was set to 10^−7^ for our study.

### 3.2. Optimized Results

The optimization problem of the moving finger is a multidimensional, nonlinear optimization problem and is solved by the inner penalty function method and Newton’s method. The changes in the target value and the convergence accuracy in the optimization iteration are presented in Figure 5, in which the iteration step is represented by *k*. The total number of steps in one iteration was 70. During the process of iteration, the value *f(x)* decrease and the convergence precision gradually met the design requirements. The minimum convergence value *ε* = 8.69 × 10^−8^, which meets the design requirements.

Table 2 lists the optimized results of the material parameters of the moving finger before and after the improvement of the inner penalty function method. The initial material was duralumin. As can be seen in the data in the table, when the unified obstacle factor of the inner point penalty function method was used to calculate the parameter *E*, the results deviated from the limits. In contrast, when the improved obstacle factor was used, the calculation results were constrained within the limits. This demonstrates that the improved internal penalty function method is valid.

## 4. The Finite Element Analysis

From the above analysis, the optimized material parameters and geometric parameters of the moving finger were obtained. The finite element analysis can figure out the vibration displacement response of the finger through simulations to verify the optimization results. In that process, the microgripper bears the dynamic load in the process of the LUMs driving in addition to its own inertia, which is very important to the stationarity and security of the microgripper.

### 4.1. Modal Analysis

Modal analysis can obtain the eigenfrequency and vibration mode of the microgripper structure, which is used to determine the vibration characteristics of the device. The microgripper model was imported in the ANSYS Workbench, and solid 187, a three-dimensional, 10-node tetrahedral solid structural unit, was adopted. To improve the calculation accuracy, the size of the mesh near the hole was set to 1 mm, and the other was set to 2 mm. The block Lanczos method was applied to extract the first six modes of the microgripper. The range of the frequency fluctuation was 246.57–559.55 Hz. Table 3 shows the eigenfrequency of the microgripper.

From the vibration modes of the microgripper, it can be the concluded that the vibrations of the microgripper are mainly concentrated on the moving finger and the fixed finger. By analyzing the harmonic response of the microgripper, which mode of vibration was excited by the LUMs can be judged.

### 4.2. Harmonic Response Analysis

Harmonic response analysis can obtain the steady-state response of the structure when it is subjected to a sinusoidal load with time. The results of the modal analysis were imported into the harmonic response analysis, and the modal superposition method was used to calculate the response of the fingers. The peak response frequency of the microgripper is shown in Table 4.

When the frequency of the exciting force is close to the above frequency points, it will have a great impact on the stability of the microgripper. For the V-type linear ultrasonic motor, the excitation frequency of piezoelectric ceramics is greater than 30 kHz; therefore, the influence of the actuator on the finger vibration can be reduced.

### 4.3. Transient Dynamics Simulation

The LUMs have a fast response. Through the friction between the driving foot and the guide rail, each stepping motion has a large output force and an instantaneous impact force, which will increase the vibration of the microgripper fingers. The displacement response of the microgripper end-effector under the instantaneous impact force of the LUMs can be obtained through transient dynamic analysis.

The ANSYS Workbench was used to perform a transient dynamic analysis of the microgripper to verify the feasibility of the optimization method. The flexure hinge, moving platform, and moving finger were analyzed as an integral structure. The structural and material parameters of the moving finger were analyzed using the initial values in Table 2 and the numerical values obtained after the algorithm was improved, respectively. The finite element model of the microgripper is shown in Figure 6.

According to the operation form of the microgripper, points B and C in the figure adopt fixed supports, and point A restricts the displacement in *x* and *y* directions. The results of finite element simulation, namely, the vibration displacement responses at the end of the moving finger, are presented in Figure 7. This figure shows that an impact force in *z* direction was applied at point A, and the vibrations of point D in *y* direction was calculated by ignoring the influence of damping. The vibration amplitude at the end of the moving finger was found to be greatly reduced after optimization, which indicates that the optimization method is feasible and effective.

## 5. Vibration Optimization Performance Test

For validating the above results of the vibration displacement response, an LK-H150 laser velocimeter was employed to test the vibration of the fingertip of the microgripper, as shown in Figure 8. The laser velocimeter was fixed on a lifting platform, the height of which could be adjusted to allow the laser to be aimed at the end-effector of the moving finger. By setting the LUMs with the same drive step and time via the controller, the initial displacement and velocity consistency of the moving finger in each group of the tests were guaranteed. The vibration data of the fingertip were collected and processed by the computer.

The moving fingers with different structures and materials were used for the vibration tests. The parameters of the materials are reported in Table 5, and the vibration amplitudes of the fingertips are shown in Figure 9.

It can be concluded from Figure 9 that, though they had the same diameter and length, the vibration amplitude of the duralumin finger was greater than that of the carbon fiber plastic finger. In other words, the smaller the density of the material and the larger the elastic modulus, the smaller the impact on the vibration of the finger. In addition, the diameter and length of the finger were found to have nonlinear effects on the vibration amplitude; the greater the diameter and the shorter the length of the fingers for all three materials, the smaller the vibration amplitude.

Figure 10 presents the tests results of the vibration displacement response of the moving finger before and after optimization. Before optimization, the diameter of the finger was 6 mm, the length was 12 mm, and the material was aluminum. The measured amplitude of the finger at the initial vibration was 16.31 μm. After optimization, the diameter was increased to 10 mm, the length was reduced to 5 mm, and the material was changed to carbon fiber plastic. The measured vibration amplitude of the finger was only 2.49 μm, which is 84.7% less than that before optimization. This proves that it is feasible to suppress the vibration of the operating end via the proposed optimization method.

## 6. Conclusions

During the operation of the microgripper, the movement of the LUM will cause the vibration of the moveable finger. The vibration characteristics of the moveable finger are determined by the geometric parameters, material parameters, and boundary conditions of the finger. Therefore, the vibration of the moveable finger can be restrained by changing the geometric and material parameters.

In this paper, the vibration equation of the movable finger of a microgripper was established based on vibration theory, and the sensitivities of the modifiable geometric and material parameters of the moveable finger were analyzed. Then, according to the analysis results, a multidimensional nonlinear optimization model was established, the target of which is the suppression of the vibration of the moveable finger, the variables of which are the geometric and material parameters, and the constraints of which are the size and material properties of the structure. The constraints were integrated into the objective function by using the improved internal penalty function method and were iteratively solved by the Newton method. The results of the obstacle factors in the internal penalty function method were compared before and after the improvement. Finally, the laser vibration measurement tests demonstrated that the vibration amplitude of the free end of the moving finger before optimization was 16.31 μm, and that after optimization was 2.49 μm, thereby exhibiting a reduction of 84.6%. This indicates that the proposed optimization method can optimize the structure and improve the positioning accuracy and stability of the microgripper.

## Figures and Tables

**Figure 1 micromachines-13-01453-f001:**
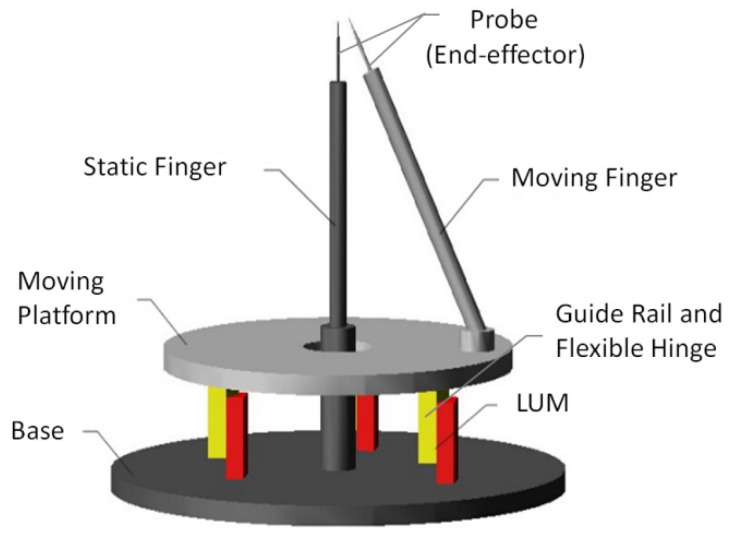
Structure of the microgripper.

**Figure 2 micromachines-13-01453-f002:**
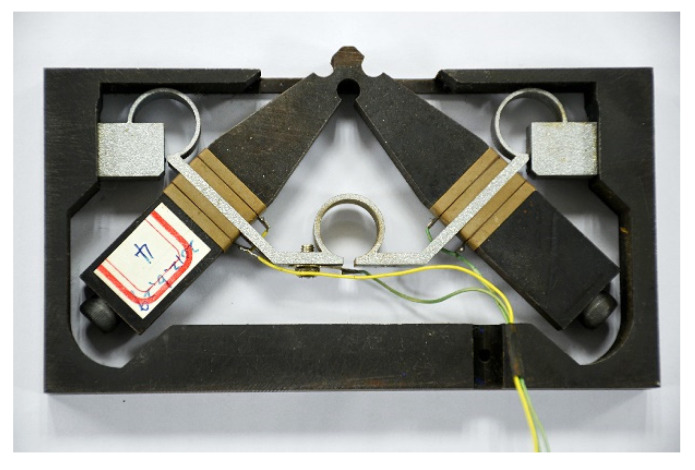
V-type linear ultrasonic motor.

**Figure 3 micromachines-13-01453-f003:**
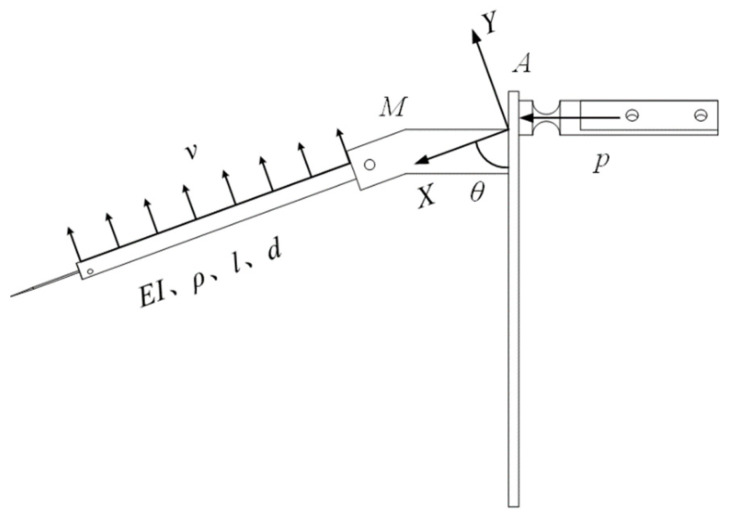
Mechanical model of the finger and circular plate.

**Figure 4 micromachines-13-01453-f004:**
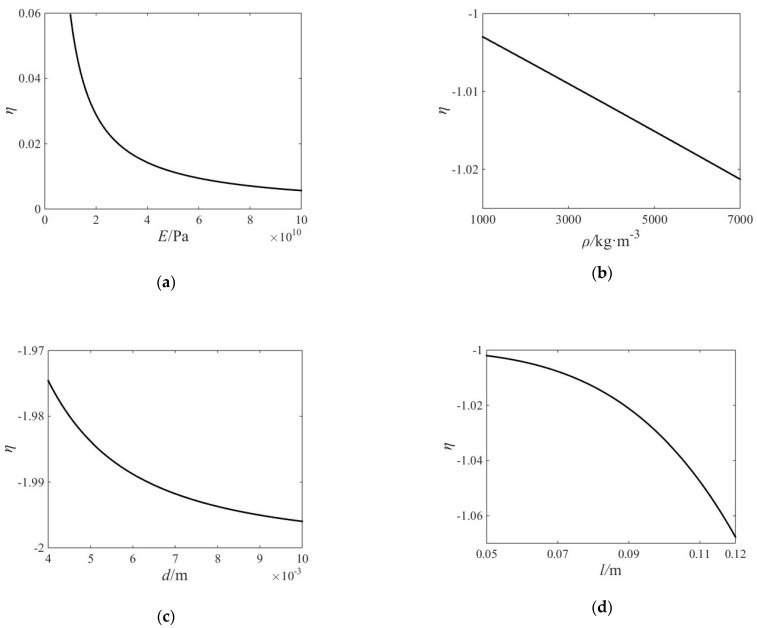
The relative sensitivity of the objective function: (**a**) sensitivity of elastic modulus *E*; (**b**) sensitivity of density *ρ*; (**c**) sensitivity of diameter *d*; (**d**) sensitivity of length *l*.

**Figure 5 micromachines-13-01453-f005:**
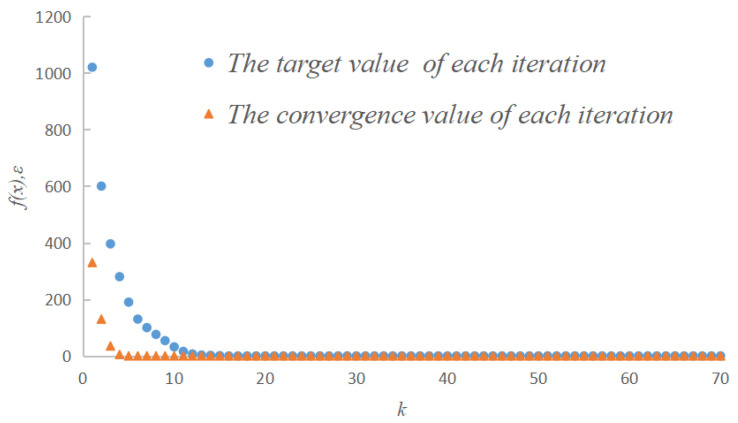
The changes in the target value and the convergence accuracy in the optimization iterations.

**Figure 6 micromachines-13-01453-f006:**
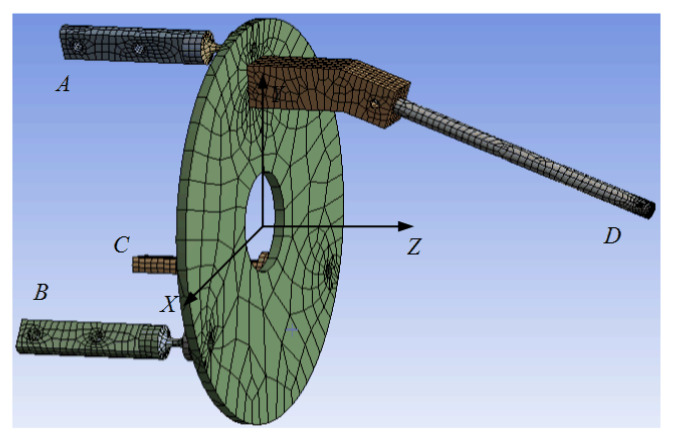
The meshed finger model.

**Figure 7 micromachines-13-01453-f007:**
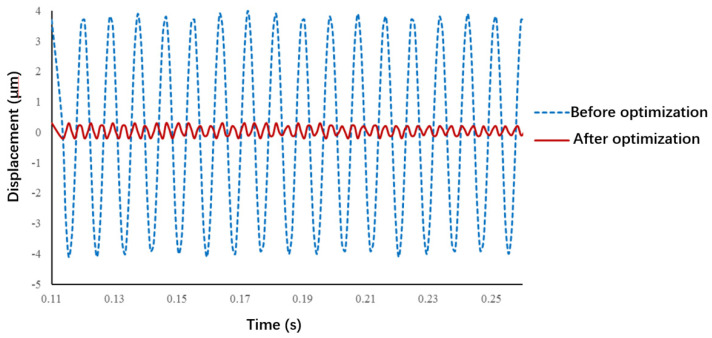
Vibration displacement response at the end of the moving finger.

**Figure 8 micromachines-13-01453-f008:**
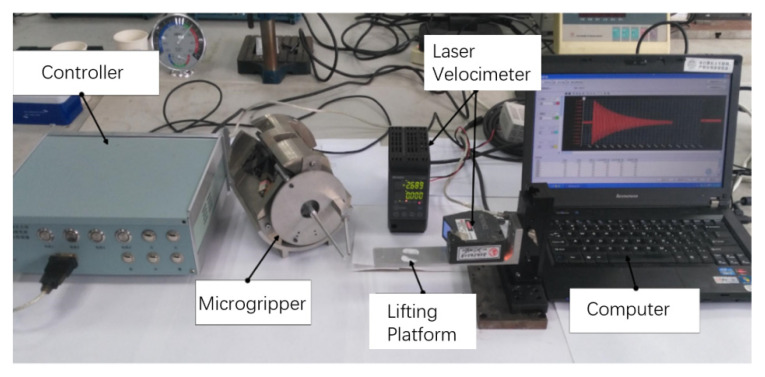
Vibration response test system.

**Figure 9 micromachines-13-01453-f009:**
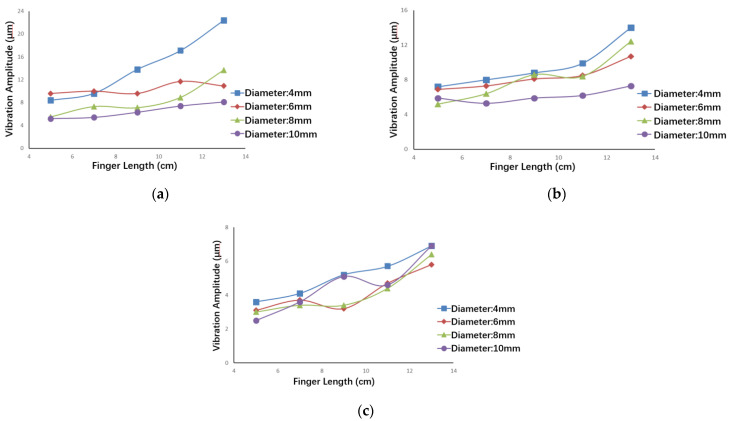
Vibration amplitudes of the fingertips: (**a**) duralumin finger; (**b**) copper finger; (**c**) carbon fiber plastic finger.

**Figure 10 micromachines-13-01453-f010:**
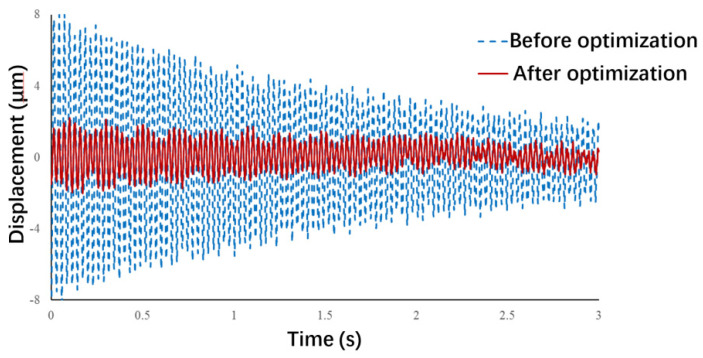
Test results of vibration displacement response.

**Table 1 micromachines-13-01453-t001:** The ranges of the structure and material parameters of the finger.

Parameters	*E*/MPa	*d*/mm	*ρ*/kg·m^−3^	*l*/mm
Value range	(1 × 10^4^, 1 × 10^5^)	(4.0, 10)	(1 × 10^3^, 7 × 10^3^)	(50, 120)

**Table 2 micromachines-13-01453-t002:** The optimized results of the moving finger.

Parameters	*E*/MPa	*d*/mm	*ρ*/kg·m^−3^	*l*/mm
Initial value	7 × 10^4^	6.0	2700	10
The value before improvement	1 × 10^4^	7.1	1552	90
The value after improvement	9.9 × 10^4^	10.0	1001	50

**Table 3 micromachines-13-01453-t003:** The eigenfrequency of the microgripper.

Order	1	2	3	4	5	6
Natural frequency/Hz	246.57	246.91	325.18	392.05	550.14	559.55

**Table 4 micromachines-13-01453-t004:** Peak response frequency.

Peak Point	1	2	3	4
Frequency/Hz	384	536	808	968

**Table 5 micromachines-13-01453-t005:** Parameters of different materials.

Materials	*E*/MPa	*ρ*/kg·m^−3^
Duralumin	7 × 10^4^	2700
Copper	1.19 × 10^5^	8900
Carbon fiber plastic	1.2 × 10^5^	1400

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
