# Peer review of "Analysis and Optimization of a Microgripper Driven by Linear Ultrasonic Motors"

_micromachines, 2022, doi:10.3390/mi13091453_

Round 1

Reviewer 1 Report

Dear Authors,

My comments and questions are listed in the attachment. 

Kind Regards

Author Response

We thank all editors and reviewer(s) for their help and effort which have contributed to improving this paper. All comments proposed by the reviewers have been carefully addressed. These comments have been very helpful to improve the manuscript. We believe that through their careful editing of the paper and the reviewer’ s insightful comments, the revised paper has been substantially improved.

Please see the attachment with our reply.

Reviewer 2 Report

(

Thanks for your peer-review invitation for the paper titled “Analysis and optimization of a microgripper driven by linear ultrasonic motors”. The end-point positioning accuracy for the microgrippers is critical to the various applications. At this point, this study is significant to positioning accuracy for precision engineering. After careful checking, some concerns about this study are provided in the following comments for further revision.

(1) Both fig 4 and fig 5 are so simple, that they may be combined.

(2) The effects of materials properties and designed structures are important. However, it is inappropriate to discuss them together as influencing factors.

(3) All parameters should be normalized in the section on sensitivity analysis, even if it is necessary to discuss the influence of both materials and structural parameters at the same time.

(4) In section 3 Structure optimization, whether the influence analysis of material parameters is suitable or not. This question should be concerned carefully by the authors.

(5) The logical connection of the work carried out in this paper has not been explained in the article. Especially for the mixed research work of analytical method, finite element method, and experimental method, if there is no strict verification relationship between them, this logical explanation is very necessary.

(6) In addition, there are several hyperlinks in the pdf text, the reason is unknown, please edit the reference. (e.g., “lower power consumption” p1 line37; “excited vibration”, p2, line50; “velocity profiles”, p2, line57)

Author Response

Thank you very much for your comments. We believe the paper has been substantially improved as a result of your comments.

Round 2

Reviewer 1 Report

Dear Authors,

You have adressed all my remarks and questions in the revision. The manuscript can be published after minor revision.

Kind Regards